# Coping strategies of COVID-19 recovered patients at the Ghana Infectious Disease Centre

Esinam Aku Amedewonu[1]*, Genevieve Cecilia Aryeetey[2]*, Anthony Godi[3], Josephine Sackeyfio[4], Alfred Dickson Dai-Kosi[4], Thomas Akuetteh Ndanu[4]

**1** Department of Anaesthesia, Korle Bu Teaching Hospital, Accra, Ghana, **2** Department of Health Policy, Planning and Management, School of Public Health, Colleges of Health Sciences, University of Ghana, Legon, Accra, Ghana, **3** Department of Biostatistics, School of Public Health, University of Ghana, Legon, Accra, Ghana, **4** Department of Community and Preventative Dentistry, University of Ghana Dental School, Korle Bu, Accra, Ghana

* eamedewonu@yahoo.com (EAA); gcaryeetey@ug.edu.gh (GCA)

## Abstract

### Background

The Coronavirus Disease (COVID-19) is a disease with diverse effects on multiple organ systems, leading to varying presentations and severe complications. As the pandemic progresses, the challenges faced by those who recovered from the disease evolved as various coping strategies were adopted post recovery.

### Aim

This study investigated the coping strategies used by individuals recovering from COVID-19 to manage the physical, psychological, and social impacts of the disease. It further explored the factors influencing these strategies and their correlation with post-recovery quality of life.

### Methods

This cross-sectional quantitative study involved 150 participants who attended the Ghana Infectious Disease Centre's post-COVID-19 review clinic between January and June 2021. Coping mechanisms were evaluated using the Brief-COPE questionnaire, which covers 28 strategies across three styles: Problem-focused coping, emotional-focused coping, and avoidant coping. Participants rated their coping strategies on a 4-point Likert scale. Analysis of variance was used to determine differences in use of coping strategies. Quality of life was assessed with the EuroQol Group Association five-domain, five-level questionnaire (EQ-5D-5L) and the EQ-VAS scale. Spearman correlation analyzed the relationship between coping strategies and quality of life.

### Results

Majority of the study participants used problem-focused (2.71 ± 0.64 SD) type of coping strategy, followed by emotional-focused coping (2.32 ± 0.43 SD). The least strategy used

**Data Availability Statement:** All relevant data are within the manuscript and its Supporting information files.

**Funding:** The author(s) received no specific funding for this work.

**Competing interests:** The authors have declared that no competing interests exist.

was avoidant coping (1.57 ± 0.39 SD). Older participants, non-healthcare workers, and those with complications or persistent symptoms exhibited higher scores in avoidant and problem-focused coping. Those with persistent symptoms had higher emotional-focused coping scores. Better quality of life was associated with less reliance on all types of coping strategies.

## Conclusion

Patients recovering from COVID-19 at the Ghana Infectious Disease Centre used positive coping mechanisms effectively. Key predictors of coping strategies included age, persistent symptoms, and complications. Improved quality of life is correlated with reduced use of coping strategies.

## Introduction

At the end of 2019, the world faced a significant public health crisis in the form of COVID-19, caused by the novel virus known as Severe Acute Respiratory Syndrome Coronavirus-2 (SARS-CoV-2) [1, 2]. Originating in Wuhan, China, this virus rapidly spread across the globe, prompting the World Health Organization to declare it a public health emergency of international concern on January 30,2020. Shortly thereafter, on March 11, 2020, it was officially declared a global pandemic, marking the first of such a declaration since the Influenza A (H1N1) pandemic in 2009. COVID-19's ability to spread quickly to over 160 countries has resulted in a staggering death toll, with over 3 million fatalities worldwide as of March 2020 [1, 3, 4].

The COVID-19 pandemic has significantly impacted various aspects of daily life and has led to disruptions in daily routines, work, education, social interactions, religious practices, and even access to basic needs like food and healthcare [5–7]. This has led to the emergence of what is now referred to as the "new normal" characterized by widespread adoption of preventive measures such as mask-wearing, social distancing, and restrictions on social gatherings, which have been implemented worldwide in various degrees of success.

Psychologically, COVID-19 has had immense negative impacts on our lives with consequences of stress, anxiety, and other mental health challenges [8–10]. Factors contributing to these challenges include the rapid transmission of the virus, high morbidity and mortality rates, and the absence of definitive treatments. Public health measures like social distancing and lockdowns have resulted in increased feelings of loneliness and isolation, exacerbating stress, fear, and anxiety.

Various studies conducted in different countries have reported elevated levels of stress, depression, anxiety, and even suicidal ideation among the general population during the pandemic [10–16]. Coping strategies are vital tools that have played a crucial role in determining mental health outcomes. These strategies can have either positive or negative effects on mental well-being, with healthy coping mechanisms reducing negative emotions and promoting resilience, while unhealthy ones exacerbate stress, potentially leading to suicidal tendencies [17].

According to Fluharty and Thai et al., there are 28 existing coping strategies categorized into '*approach coping*' and '*avoidant coping*' [18, 19]. Approach coping involves actively addressing stressors through actions such as seeking emotional support, planning, and positive reframing. In contrast, avoidant coping seeks to avoid stressors and may include denial and

substance use. Other categories such as '*emotional-focused*' and '*problem-focused*' strategies, have also been used to classify coping mechanisms. The choice of coping style strategies can be influenced by factors such as gender, personality, socio-economic position, psycho-social factors, and socio-demographic factors.

The ways in which individuals deal with stressors can have long-term health implications [20]. Anxiety and depression typically result from avoidance coping strategies. Cognitive behavioural therapy has been shown to be the most effective therapy for targeting avoidance strategies and this focuses on cognitive reappraisal and problem-solving responses. Understanding the patterns of coping strategies and their predictors is essential for selecting appropriate treatment modalities and providing psychological support to individuals experiencing pandemic-related stressors.

The study thus sought to identify the coping strategies utilized by COVID-19 recovered patients in Ghana, assess the association of socio-demographic characteristics with coping strategies used by these patients, and determine the relationship between quality of life and coping strategies employed by COVID-19 recovered individuals.

## Materials and methods

### Study setting

The study was conducted at the Ghana Infectious Disease Center (GIDC) post-COVID-19 review clinic. The Ghana Infectious Disease Center, the nation's inaugural infectious disease facility with a capacity of 100 beds, was established in response to the COVID-19 pandemic. Within this center, a post- COVID-19 clinic was established to provide follow-up care for all individuals who had recovered from COVID-19. This clinic typically attended to approximately five patients per day. The initial follow-up appointment took place approximately one to two weeks after a patient's discharge, and most patients underwent two to three additional reviews over a six-week period. Approximately 80% of the patients who were previously hospitalized or isolated at the GIDC returned for follow-up at the clinic. The primary focus of the clinic is to assess and manage any lingering COVID-19 symptoms, as well as to address any exacerbation of pre-existing co-morbid conditions that may have occurred due to COVID-19.

### Study population, sampling and data collection

The study included all recovered patients aged 18 years and above who attended the Ghana Infectious Disease Centre (GIDC) post-COVID-19 review clinic between August 2021 and November 2021. These patients were willing and able to provide informed consent for participation.

From January 2021 to June 2021, 782 COVID-19 patients attended the GIDC COVID-19 clinic. Following recovery, 209 of them were officially discharged to the post-COVID-19 clinic. These 209 patients formed the study population. Using Yamane's formula, a minimum sample size of 137 patients was estimated with a precision level of 5%. Considering a 10% attrition rate (13 patients), the final sample size was determined to be 150 patients. Purposive sampling was employed to recruit study participants. Clinical and biographic data were extracted from patients' clinical records, including information related to their first COVID-19 positive test, negative test, length of hospital or isolation center stay, and comorbidities.

Coping was measured using the Brief-COPE questionnaire as developed in 1989 by Carver [21]. This consists of 14 subscales comprising 2 items each, making a total of 28 coping strategies (28 different questions which depict a particular coping strategy). These are classified under three overarching styles of coping [18, 22]. The first strategy is 'Problem-Focused Coping' which includes Active Coping, Positive Reframing, Planning, and the Use of

Informational Support. This first strategy is a positive outcome strategy which aims at changing the stressful event. The second strategy is 'Emotional-Focused Coping' which includes the Use of Emotional Support, Acceptance, Humour, Self-Blame, Religion, and Venting. This has both positive and negative outcomes and aims at regulating emotions associated with stressors. The third strategy is 'Avoidant Coping' which includes Self-Distraction, Behavioural Disengagement, Substance Use, and Denial. This third strategy is a negative outcome strategy. It is generally maladaptive and harmful to one's well-being. High scores are indicative of efforts to disengage physically or cognitively from the stressful situation whereas low scores indicate adaptive coping. Participants were asked to rate the degree to which they used each coping strategy to deal with the stressful event of COVID-19 during their recovery.

Other demographic information was obtained through the questionnaires. Data collection and extraction of information from the clinical records, started on 13th August 2021 and ended on 18th November 2021.

## Data analysis

The Brief-COPE questionnaire consists of 14 subscales comprising 2 items each, making a total of 28 coping strategies (28 different questions which depict a particular coping strategy). These are classified under three overarching styles of coping [18, 22]. The first strategy is '*Problem-Focused Coping'* which is a positive outcome strategy. The second strategy is '*Emotional-Focused Coping'* which has both positive and negative outcomes and aims at regulating emotions associated with stressors. The third strategy is '*Avoidant Coping'*. This is a negative outcome strategy and is generally maladaptive and harmful to one's well-being. High scores are indicative of efforts to disengage physically or cognitively from the stressful situation whereas low scores indicate adaptive coping. Participants were asked to rate the degree to which they used each coping strategy to deal with the stressful event of COVID-19 during their recovery.

The Brief-COPE used in this study to assess coping strategies has been validated in several populations, has different language translations and has a good Cronbach's alpha of 0.87 [23–25] which was comparable to the Cronbach's alpha reliability coefficient in this study which was 0.82.

A 4-point Likert scale was used for rating coping strategies, and this ranged from 1 –lowest frequency to 4 –highest frequency in each coping activity/strategy. The coping strategies were grouped into three broad themes and each theme further broken down into sub-themes made up of pairs of similar strategies. The average scores for the paired coping strategies as well as for the three overarching sub-themes were computed and presented to indicate the degree to which study participants utilized each coping style.

The EQ-5D-5L and EQ-VAS tools were used to measure QoL scores. The EQ-5D-5L tool has five domains (mobility, self-care, usual activities, pain/discomfort, and anxiety/depression) with five levels each, ranging from 1 (best) to 5 (worst).

Participants also rated their health on the EQ-VAS, a scale from 0 to 100, where 0 means "worst imaginable health" and 100 means "best imaginable health status".

One-way analysis of variance (ANOVA) was used to test for differences in means across categories of the socio-demographic factors, symptoms, comorbidities, complications, and treatment modalities. A level of significance was set at $p < 0.05$. Graphs of means and standard deviations were used to compare scores of the different types of coping strategies used. Spearman correlation was used to establish a linear relationship between quality of life and Coping strategies. Internal consistency of the tools used in measuring coping strategies and quality of life was assessed using the Cronbach's alpha.

### Ethical considerations

The research proposal was submitted to the Ghana Health Service Ethics Review Committee to attain ethical approval to undertake the study (Approval ID: GHS–ERC 029/07/21). A permission letter was also provided by the School of Public Health to the Ghana Infectious Disease Centre (GIDC) for study site approval. Informed consent was obtained from each participant through a written consent form attached to each questionnaire.

## Results

### Background characteristics

From Table 1, the average age of the participants was 43 ± 15 years, with the largest proportion (35.5%) falling within the 30–39 year group. Females constituted the majority of the recovered

**Table 1. Demographic characteristics of study respondents (N = 150).**

| | N | % |
|---|---|---|
| Age (years), Mean = 43.3, SD = 14.8 | | |
| <30 | 21 | 14.0 |
| 30–39 | 53 | 35.3 |
| 40–49 | 30 | 20.0 |
| 50–59 | 23 | 15.3 |
| 60+ | 23 | 15.3 |
| Sex | | |
| Male | 68 | 45.3 |
| Female | 82 | 54.7 |
| Religion | | |
| Christian | 135 | 90.0 |
| Muslim | 8 | 5.3 |
| Other | 7 | 4.7 |
| Ethnicity | | |
| Ga-Dangme | 32 | 21.3 |
| Akan | 69 | 46.0 |
| Ewe | 29 | 19.3 |
| Northern tribe | 12 | 8.0 |
| Other | 8 | 5.3 |
| Place of residence | | |
| Urban | 142 | 94.7 |
| Non-urban | 8 | 5.3 |
| Educational level | | |
| Primary | 5 | 3.3 |
| Secondary | 11 | 7.3 |
| Tertiary | 125 | 83.3 |
| Vocational/Technical | 9 | 6.0 |
| Occupation | | |
| Student | 3 | 2.0 |
| Formally employed | 107 | 71.3 |
| Informally employed | 23 | 15.3 |
| Unemployed | 8 | 5.3 |
| Retired | 9 | 6.0 |
| Total | 150 | 100.0 |

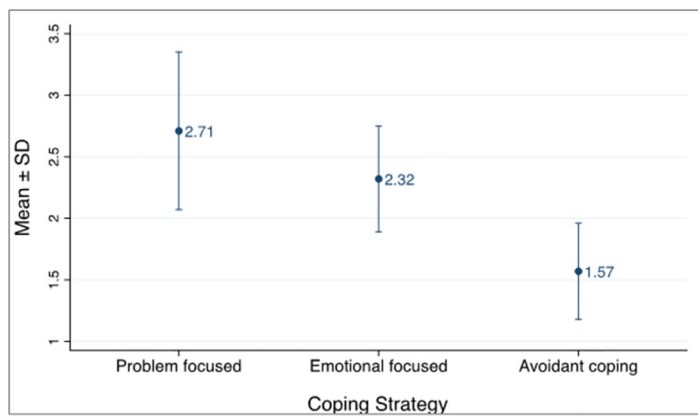

**Fig 1. Overall coping strategy.**

patients (54.7%), and the prevalent religion among them was Christianity (90%). A significant portion (46%) identified as primarily Akan for ethnicity. The vast majority (94.7%) resided in urban areas. Furthermore, 83.3% had achieved tertiary level education, and 71.3% were formally employed.

## Coping strategies used by study participants

The highest coping strategy used by participants was problem-focused coping with a mean of 2.71 ± 0.64 SD on a scale 1 (no use at all) to 4 (used a lot) as seen in Fig 1. This was followed by emotional-focused coping with a mean of 2.32 ± 0.43 SD and lastly avoidant coping with a mean of 1.57 ± 0.39 SD.

## Problem-focused coping

With the specific type of problem-focused coping, majority of respondents reported the use of informational support as their predominant coping style with a mean of 3.07 ± 0.98 SD as shown in Fig 2. The least used was positive reframing with a mean of 2.23 ± 1.00 SD.

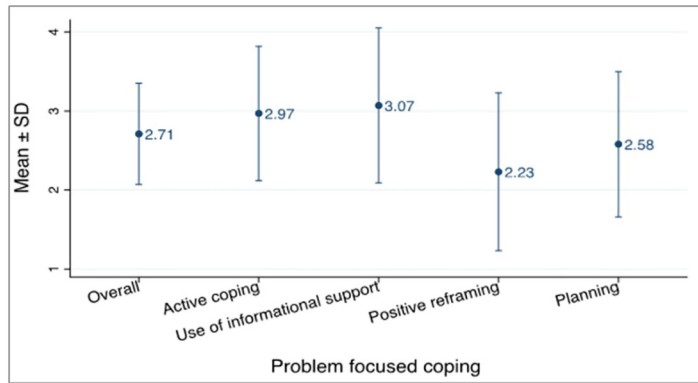

**Fig 2. Problem-focused coping strategy.**

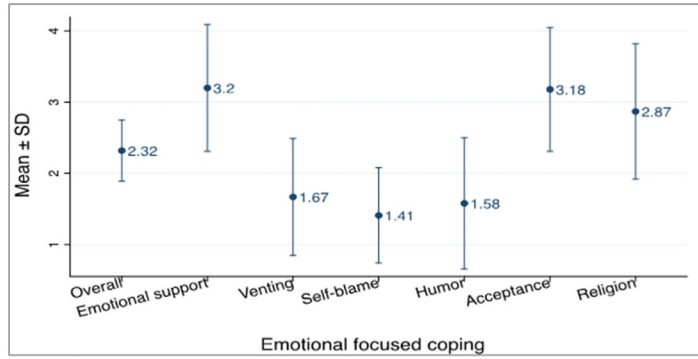

**Fig 3. Emotional-focused coping strategy.**

## Emotional-focused coping

With Emotional-focused coping strategy, majority of respondents utilised emotional support as their main coping style with a mean of 3.20 ± 0.89 SD as shown in Fig 3. The least used style of coping was self-blame with a mean of 1.41 ± 0.67 SD.

## Avoidant coping

With Avoidant coping, the most used coping style was self-distraction with a mean of 2.57 ± 1.03 SD and the least reported style of coping was substance use with a mean of 1.05 ± 0.20 SD as shown in Fig 4.

## Predictors of coping strategies used by study participants

**Socio-demographic characteristics and coping strategies.** Significant differences in mean avoidant coping scores were observed across age categories (p = 0.011) where the highest score was observed among those aged 40–49 years and the least among those aged 30–39 years as seen in Table 2. There were also significant differences in avoidant coping scores (p = 0.004) as well as problem-focused coping scores (p = 0.013) between healthcare and non-healthcare workers where higher scores were observed for the latter. There were no significant differences observed for the other socio-demographic characteristics (sex, religion, ethnicity, marital status, place of residence and occupation) across all three coping strategy styles.

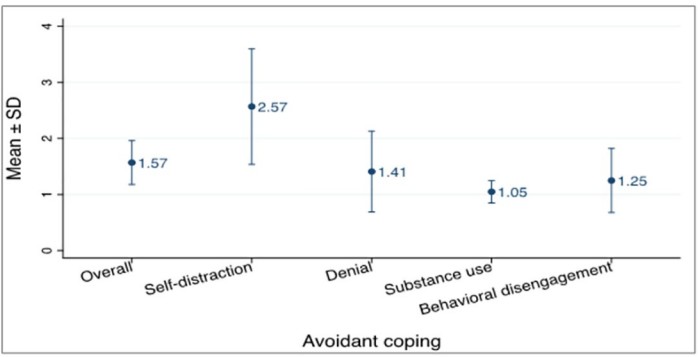

**Fig 4. Avoidant coping strategy.**

**Table 2. Socio-demographic characteristics and coping strategies.**

| | Avoidant coping | | | Problem-focused | | | Emotional-focused | | |
|---|---|---|---|---|---|---|---|---|---|
| | **Mean** | **SD** | **P-value** | **Mean** | **SD** | **P-value** | **Mean** | **SD** | **P-value** |
| Age (years) | | | 0.011 | | | 0.618 | | | 0.623 |
| <30 | 1.63 | 0.39 | | 2.55 | 0.56 | | 2.33 | 0.35 | |
| 30–39 | 1.43 | 0.28 | | 2.69 | 0.67 | | 2.33 | 0.41 | |
| 40–49 | 1.70 | 0.43 | | 2.76 | 0.63 | | 2.37 | 0.47 | |
| 50–59 | 1.68 | 0.39 | | 2.85 | 0.69 | | 2.35 | 0.50 | |
| 60+ | 1.55 | 0.44 | | 2.70 | 0.63 | | 2.19 | 0.46 | |
| Sex | | | 0.117 | | | 0.394 | | | 0.094 |
| Male | 1.52 | 0.39 | | 2.66 | 0.65 | | 2.25 | 0.39 | |
| Female | 1.62 | 0.38 | | 2.75 | 0.64 | | 2.37 | 0.46 | |
| Religion | | | 0.750 | | | 0.815 | | | 0.208 |
| Christian | 1.56 | 0.38 | | 2.70 | 0.65 | | 2.32 | 0.44 | |
| Muslim | 1.67 | 0.55 | | 2.84 | 0.64 | | 2.48 | 0.29 | |
| Other | 1.57 | 0.26 | | 2.75 | 0.49 | | 2.08 | 0.35 | |
| Ethnicity | | | 0.337 | | | 0.717 | | | 0.233 |
| Ga-Dangme | 1.50 | 0.27 | | 2.81 | 0.67 | | 2.26 | 0.51 | |
| Akan | 1.61 | 0.39 | | 2.73 | 0.64 | | 2.38 | 0.40 | |
| Ewe | 1.49 | 0.33 | | 2.58 | 0.62 | | 2.18 | 0.46 | |
| Northern tribe | 1.67 | 0.64 | | 2.69 | 0.58 | | 2.40 | 0.34 | |
| Other | 1.69 | 0.41 | | 2.64 | 0.81 | | 2.40 | 0.41 | |
| Marital status | | | 0.560 | | | 0.183 | | | 0.324 |
| Single | 1.57 | 0.35 | | 2.75 | 0.61 | | 2.37 | 0.31 | |
| Married | 1.57 | 0.39 | | 2.69 | 0.64 | | 2.31 | 0.46 | |
| Separated/ Divorced | 1.88 | 0.76 | | 3.46 | 0.52 | | 2.64 | 0.57 | |
| Widow(er) | 1.53 | 0.36 | | 2.59 | 0.71 | | 2.19 | 0.43 | |
| Place of residence | | | 0.956 | | | 0.337 | | | 0.917 |
| Urban | 1.57 | 0.38 | | 2.70 | 0.65 | | 2.32 | 0.44 | |
| Non-urban | 1.58 | 0.45 | | 2.92 | 0.53 | | 2.30 | 0.27 | |
| Educational level | | | 0.136 | | | 0.689 | | | 0.424 |
| Primary | 1.85 | 0.32 | | 2.93 | 0.83 | | 2.22 | 0.73 | |
| Secondary | 1.50 | 0.46 | | 2.83 | 0.80 | | 2.42 | 0.42 | |
| Tertiary | 1.55 | 0.36 | | 2.70 | 0.64 | | 2.30 | 0.43 | |
| Vocational/Technical | 1.76 | 0.61 | | 2.56 | 0.43 | | 2.51 | 0.34 | |
| Occupation | | | 0.213 | | | 0.469 | | | 0.478 |
| Student | 1.75 | 0.13 | | 2.58 | 0.63 | | 2.28 | 0.32 | |
| Formally employed | 1.53 | 0.34 | | 2.68 | 0.63 | | 2.32 | 0.45 | |
| Informally employed | 1.67 | 0.53 | | 2.92 | 0.68 | | 2.43 | 0.38 | |
| Unemployed | 1.78 | 0.34 | | 2.50 | 0.65 | | 2.14 | 0.53 | |
| Retired | 1.58 | 0.47 | | 2.71 | 0.75 | | 2.21 | 0.36 | |
| Healthcare worker | | | 0.004 | | | 0.013 | | | 0.648 |
| No | 1.65 | 0.39 | | 2.82 | 0.61 | | 2.33 | 0.42 | |
| Yes | 1.46 | 0.36 | | 2.55 | 0.66 | | 2.30 | 0.46 | |
| Overall | 1.57 | 0.39 | | 2.71 | 0.64 | | 2.32 | 0.43 | |

**Table 3. COVID-19 related factors and coping strategies.**

| | Avoidant coping | | | Problem-focused | | | Emotional-focused | | |
|---|---|---|---|---|---|---|---|---|---|
| | Mean | SD | P-value | Mean | SD | P-value | Mean | SD | P-value |
| Persistent symptoms (after 30 days) | | | 0.135 | | | 0.242 | | | 0.031 |
| No | 1.52 | 0.30 | | 2.64 | 0.68 | | 2.24 | 0.47 | |
| Yes | 1.62 | 0.45 | | 2.77 | 0.61 | | 2.39 | 0.39 | |
| Comorbidities prior to COVID-19 infection | | | 0.233 | | | 0.710 | | | 0.869 |
| No | 1.54 | 0.34 | | 2.69 | 0.64 | | 2.32 | 0.43 | |
| Yes | 1.62 | 0.43 | | 2.73 | 0.64 | | 2.31 | 0.44 | |
| Comorbidities worsening post-COVID-19 infection | | | 0.051 | | | 0.589 | | | 0.840 |
| No | 1.54 | 0.34 | | 2.69 | 0.66 | | 2.32 | 0.42 | |
| Yes | 1.68 | 0.49 | | 2.76 | 0.60 | | 2.30 | 0.47 | |
| Complications post-COVID-19 infection | | | <0.001 | | | 0.037 | | | 0.061 |
| No | 1.51 | 0.33 | | 2.65 | 0.63 | | 2.28 | 0.44 | |
| Yes | 1.78 | 0.47 | | 2.91 | 0.63 | | 2.44 | 0.41 | |
| Overall | 1.57 | 0.39 | | 2.71 | 0.64 | | 2.32 | 0.43 | |

**COVID-19 related factors and coping strategies.** Significantly higher scores were observed in both Avoidant coping (p<0.001) and Problem-focused coping (p = 0.037) for participants who developed complications post-COVID-19 infection compared to those who did not develop any complications as seen in Table 3. Participants who had persistent symptoms after 30 days also had higher Emotional-focused coping scores than those who didn't have such symptoms (p = 0.031). No significant differences in coping strategy scores were observed for the other COVID-19 related factors (prior comorbidities and worsening comorbidities) strategies.

**Association between quality of life and coping strategies.** From the Spearman correlation analysis, higher quality of life EQ-5D-5L scores were significantly associated with lower use of positive reframing, venting, religion, and behavioural disengagement coping strategies as seen in Table 4. Higher quality of life EQ-VAS scores were also significantly associated with lower use of informational support, venting, and behavioural disengagement coping strategies. Overall, higher EQ-5D-5L quality of life was significantly associated with lower use of emotional-focused coping strategies while higher EQ-VAS quality of life was significantly associated with lower use of both problem-focused and avoidant coping strategies.

## Discussion

The coping strategy types utilized the most by the study participants were problem-focused with a mean score of 2.71 ± 0.64 SD, followed by emotional-focused coping with a mean score of 2.32 ± 0.43 SD. The least used strategy types were avoidant coping with a mean score of 1.57 ± 0.39 SD. Problem-focused coping is a positive coping strategy aimed at changing stressful events [26]. Emotional-focused coping has both positive and negative strategies, but generally more positive outcomes aimed at regulating emotions associated with stressors. Avoidant coping is generally a negative strategy with harmful effects to one's physical well-being due to lack of actions to reduce stressors resulting in self-blame and helplessness [19]. If used in the short-term period however avoidant coping may be considered positive in helping reduce acute stress.

Findings from this study suggests that participants coped well during the COVID-19 pandemic using mostly positive coping mechanisms. Being a novel disease and causing havoc on

**Table 4. Coping strategies and quality of life.**

| Coping strategy | EQ-5D-5L | | EQ-VAS | |
|---|---|---|---|---|
| | Rho | P-value | Rho | P-value |
| **Problem-focused** | -0.154 | 0.059 | -0.167 | 0.042 |
| Active coping | 0.098 | 0.235 | -0.021 | 0.802 |
| Use of informational support | -0.062 | 0.455 | -0.203 | 0.013 |
| Positive reframing | -0.279 | 0.001 | -0.102 | 0.214 |
| Planning | -0.151 | 0.065 | -0.134 | 0.101 |
| **Emotional-focused** | -0.243 | 0.003 | -0.1 | 0.223 |
| Emotional support | -0.024 | 0.771 | -0.1 | 0.224 |
| Venting | -0.419 | <0.001 | -0.245 | 0.003 |
| Self-blame | -0.074 | 0.371 | -0.115 | 0.162 |
| Humour | 0.008 | 0.921 | 0.123 | 0.134 |
| Acceptance | -0.049 | 0.552 | 0.067 | 0.415 |
| Religion | -0.166 | 0.042 | -0.091 | 0.268 |
| **Avoidant** | -0.042 | 0.606 | -0.228 | 0.005 |
| Self-distraction | 0.108 | 0.188 | -0.06 | 0.466 |
| Denial | -0.038 | 0.645 | -0.142 | 0.083 |
| Substance use | 0.105 | 0.199 | -0.063 | 0.446 |
| Behavioural disengagement | -0.428 | <0.001 | -0.302 | <0.001 |

the global front, many media outlets both locally and internationally provided daily updates on the virus. This was consistent with this study's findings where use of informational support was the most used problem-focused strategy with a mean score of 3.07 ± 0.98 SD.

A study conducted in Vietnam by Thai et al. [18] among public health and preventive medicine students on perceived stress and coping strategies during the COVID-19 pandemic reported on the predominant use of approach coping strategies (active coping, acceptance, emotional support, planning, informational support, positive reframing, and planning) and this concurred with the strategies used by participants in this study. This may probably be due to the massive wealth of information that was being released daily about the virus leading to improved knowledge, awareness, and the necessary skills for coping although it was done in different target populations.

Religion is a major aspect of the lives of many Ghanaians. A high number of respondents used religion as a coping strategy with a mean score of 2.87 ± 0.95 SD. Studies conducted in some countries such as Columbia reported on religion having a potential impact on coping [27, 28]. The reason for this could be from fear and panic which was evident in the early stages of the disease when not much was known about the virus. Many churches made use of online platforms to organize religious services and activities forming a psychosocial support system aiding coping mechanisms [29].

Self-distraction was the most used form of avoidant coping mechanism with a mean score of 2.57 ± 1.03 SD. The acute use of strategies such as watching T.V, reading, listening to music, and sleeping as ways of distracting oneself during a crisis has been reported as helpful [30]. A possible explanation could be that the mind is taken off the stressor acutely, thereby reducing the stress. Substance use was the least used coping mechanism with a mean score of 1.05 ± 0.20 SD. Participants reported resorting to the use of alcoholic beverages as a means of reducing the stressor and this was confirmed in studies done in Ghana by Asiamah et al. [31] who reported a rise in risky health behaviours during the pandemic. Contrary to these findings were studies

conducted in Spain reporting reductions in such behaviours due to the stricter and relatively longer lockdown period [32].

### Predictors of coping strategies

Middle aged adults (40–49 years) were found to use more of avoidant coping strategies, and this was found to be statistically significant. They were also found to use more emotional-focused strategies whilst older adults aged 50–59 years used more of problem-focused coping mechanisms. This was similar to findings from Fluharty et al. [19] who reported older adults (30–59 years) and the elderly (60+ years) using more of problem-focused and emotional-focused coping. In our study, the coping strategy for those aged 40–49 years was more focused on avoidant and emotional coping mechanisms whiles those aged 50–59 years was more focused on problem coping mechanisms. It however differed from findings in that same study where avoidant coping was found to rather be used more by younger adults aged 18–29 years. From literature, older adults are more likely to use approach coping strategies and less likely to adopt avoidant coping strategies owing to years of experience in dealing with stressors [33, 34].

Females were noticed to have used more of all three coping mechanisms than men in this study. Although this was not found to be statistically significant, it concurred with findings from Fluharty et al. [19] who also reported women significantly using more of problem-focused, emotional-focused, and avoidant coping strategies than men. Several studies have also confirmed sex being a demographic predictor of coping strategies with women scoring higher than men across all ranges of styles of coping [35, 36]. A study conducted in Spain concluded on men being psychologically more affected during the lockdown [37]. Men being mostly bread winners, and the lack of income as the result of the lockdown may increase the depression and anxiety, resulting in their ability to cope well.

This study recorded participants with the least level of education using more of avoidant coping which corresponded to other research which also found avoidant coping being used more in people with lower educational levels [19]. In other studies, participants with a higher level of education used more of problem-focused mechanisms. In this study however, it was the participants with lower educational levels who mostly used problem-focused coping mechanisms. A possible reason for people with higher educational levels using more active coping mechanisms could be their exposure to better problem-focused strategies during their years of studies.

Problem-focused and emotional-focused coping mechanisms was common among people who were informally employed whilst avoidant coping was common among those unemployed. This was consistent with reports from other studies which also found active coping mechanisms being more commonly used among those who were employed [19]. In contrast to these findings, other studies have reported individuals on self-employment (informally employed) as less likely to cope well during the pandemic [29]. This is probably because they do not earn monthly salary therefore the lockdown affected their businesses and income [38, 39].

This current study showed that participants who were separated/divorced used all three coping strategies more although this finding was not statistically significant. This was consistent with literature reporting positive coping strategies being more commonly used among those separated/divorced [29]. An explanation for this may be because due to the divorce, there is likely to be more social support from family members to cope with the COVID-19. The divorce could also have been a stressor they had dealt with prior to the COVID-19 resulting in a better likelihood of coping better during the pandemic.

Participants living in non-urban places were recorded to use more of avoidant and problem-focused coping strategies whilst those living in urban areas used more of emotional-focused coping mechanisms. Studies by Fluharty et al. [19] similarly reported their respondents living in urban areas used more of supportive coping strategies. This may be because urban areas are more densely populated [40]. The urban lifestyle is more stressful hence people may have already developed certain coping strategies for survival. They may therefore be better suited to cope with the COVID-19 than those living in the non-urban areas.

Non-health workers used all three coping strategies more than healthcare workers which was found to be statistically significant. A possible reason for healthcare workers coping less effectively could be because of the large scale of morbidity and mortality they had to observe physically and deal with daily at the workplace which could have been a huge stressor during the pandemic. Literature suggests that healthcare workers are faced with higher levels of both psychological and physical stress resulting from their working environment [40]. Hence the pandemic imposed extra stress on health workers which affected their ability to cope well.

People who developed complications due to COVID-19 were associated with greater use of all three coping strategies than those who did not get any complications with significant associations observed for both avoidant and problem-focused coping styles. These findings were consistent with studies conducted by Fluharty et al. [19] who also reported people suffering from adversities as a result of COVID-19 using more of avoidant coping. It however differed from that same study which reported participants with complications using less of emotional-focused and problem-focused coping mechanisms than those who didn't have complications.

Having comorbidities prior to getting COVID-19 and having comorbidities worsening during the infection seemed to be associated with more use of avoidant coping and problem-focused strategies but less use of emotional-focused coping. These were however not statistically significant in this study. Other studies have also reported similar findings where having comorbidities prior to COVID-19 was shown to decrease one's chances of coping well [29].

Participants with persistent symptoms lasting more than 30 days commonly used all three coping strategies more than those who didn't have persistent symptoms, and this was found to be significant for only the emotional-focused strategy.

## Association between quality of life and coping strategies

When people are faced with stressful situations, they use a coping strategy to help them overcome the stress. Many studies have reported adaptive or approach coping mechanisms being associated with better quality of life [41, 42]. A reduction in the stressor from effective coping results in the improvement in quality of life with a subsequent resultant decrease in the coping mechanisms being used for the stressor.

In general, the need for the use of coping skills occurs more when QoL is low. In this study, an increase in the EQ-5D-5L quality of life of participants was generally associated with a decrease in the overall need to use a coping strategy as seen from the correlation analysis and this was statistically significant for emotional-focused coping strategies. A possible reason for this finding could be because as the stressors in the lives of participants reduced, their quality-of-life improved leading to a gradual decrease in their need to use a coping mechanism. With more information being discovered on the novel virus, better and more effective treatment modalities are being instituted, resulting in a decrease in the fear and stigma surrounding the disease. The increased knowledge and understanding of the stressor (COVID-19) has resulted in a reduction in morbidity and mortality globally, with more people learning to cope better. People may only find the necessity to cope when there are ongoing stressors.

This study also detected a somewhat positive association between improvement in EQ-VAS quality of life of respondents and their use of humour for coping. They generally laughed more and made more jokes as their condition improved. They also used more active coping techniques as their quality-of-life improved allowing them to cope better during the pandemic. This was corroborated by other studies which also found a positive association between quality of life and positive coping strategies [43].

Participants in this study also acutely used more self-distraction methods like watching more TV and reading as their quality of life improved. An explanation for this may be due to these habits being nurtured during the lockdown period hence becoming a part and parcel of their new lifestyle. They however developed some negative coping mechanisms of using substances like alcohol more during the period as life got better probably to enable them to forget about the difficulties they faced, the trauma they experienced or to celebrate surviving the disease.

Some limitations of the study are the potential for recall bias to measure the coping strategies of participants which accuracy of the information provided. It also did not compare their situation before and after COVID-19, so it could not show causal effects. Lastly, the study was limited to one treatment centre and may not represent other patients' experiences. The findings may change as research on COVID-19 evolves.

## Conclusion

During the pandemic, COVID-19 patients generally coped very well using mostly positive coping strategies. Socio-demographic factors such as age and being anon-healthcare worker significantly predicted some of the coping strategies that were used. Other significant predictors included persistent symptoms after 30 days and developing complications associated with COVID-19.

An increasing EQ-5D-5L Quality of life was associated with a decreasing need to use majority of the coping strategies. It was however positively associated with active coping, humour, and substance use. Rehabilitation (especially psychotherapy / clinical psychology) has been shown to considerably improve these factors. Improvement of these factors together with the adaptation of positive coping mechanisms results in an overall improvement in quality of life.

This study hopes to contribute valuable insights into the adaptive responses of COVID-19 survivors and provides guidance for healthcare professionals, policymakers, and support systems aiming to improve the post-COVID-19 experience for individuals and communities.

## Supporting information

**S1 Checklist. Human participants research checklist.**
(DOCX)

**S1 Dataset.**
(XLS)

## Acknowledgments

1. Levi Nii Ayi Ankrah, *National Reconstructive Plastic Surgery & Burns Centre*, *Korle Bu Teaching Hospital*, *Accra*, *Ghana*

2. Dr. Oliver-Commey and all the staff of Ghana Infectious Disease Centre (GIDC)

3. All the patients of the post-COVID-19 review clinic.

## Author Contributions

**Conceptualization:** Esinam Aku Amedewonu, Genevieve Cecilia Aryeetey.

**Data curation:** Esinam Aku Amedewonu, Anthony Godi.

**Formal analysis:** Esinam Aku Amedewonu, Genevieve Cecilia Aryeetey, Anthony Godi.

**Methodology:** Esinam Aku Amedewonu, Genevieve Cecilia Aryeetey, Anthony Godi.

**Resources:** Esinam Aku Amedewonu.

**Supervision:** Genevieve Cecilia Aryeetey.

**Validation:** Esinam Aku Amedewonu, Genevieve Cecilia Aryeetey, Anthony Godi, Thomas Akuetteh Ndanu.

**Visualization:** Esinam Aku Amedewonu, Genevieve Cecilia Aryeetey, Anthony Godi, Josephine Sackeyfio, Alfred Dickson Dai-Kosi.

**Writing – original draft:** Esinam Aku Amedewonu, Genevieve Cecilia Aryeetey, Anthony Godi, Josephine Sackeyfio, Thomas Akuetteh Ndanu.

**Writing – review & editing:** Esinam Aku Amedewonu, Genevieve Cecilia Aryeetey, Anthony Godi, Josephine Sackeyfio, Alfred Dickson Dai-Kosi, Thomas Akuetteh Ndanu.

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
