## [Decision Letter · Decision Letter 0]

12 Aug 2024

PONE-D-24-14288Coping Strategies of COVID-19 recovered patients at the Ghana Infectious Disease CentrePLOS ONE

Dear Dr. Aryeetey,

Thank you for submitting your manuscript to PLOS ONE. After careful consideration, we feel that it has merit but does not fully meet PLOS ONE’s publication criteria as it currently stands. Therefore, we invite you to submit a revised version of the manuscript that addresses the points raised during the review process.

We look forward to receiving your revised manuscript.

Kind regards,

Anthony Mwinilanaa Tampah-Naah

Academic Editor

PLOS ONE

Journal Requirements:

2. We noted in your submission details that a portion of your manuscript may have been presented or published elsewhere. “This is the second manuscript in line with our study on quality of life and coping strategies of COVID-19 recovered patients in Ghana. We have submitted revisions to the first paper i.e. quality of life to this same journal and it is under review. However the analysis conducted here is not a duplication of the previous paper. We only used some of the results or concepts, particularly quality of life to determine the relationship between  coping strategies and quality of life.”

Reviewers' comments:

Reviewer's Responses to Questions

**Comments to the Author**

1. Is the manuscript technically sound, and do the data support the conclusions?

Reviewer #1: Partly

Reviewer #2: Yes

2. Has the statistical analysis been performed appropriately and rigorously? 

Reviewer #1: Yes

Reviewer #2: Yes

3. Have the authors made all data underlying the findings in their manuscript fully available?

Reviewer #1: Yes

Reviewer #2: Yes

4. Is the manuscript presented in an intelligible fashion and written in standard English?

Reviewer #1: No

Reviewer #2: Yes

5. Review Comments to the Author

Reviewer #1: The topic is interesting, however, the study has some uncertain points and requires major revisions in order to meet the journal’s requirements

• Post COVID-19 condition, commonly known as long COVID is defined as the continuation or development of new symptoms 3 months after the initial SARS-CoV-2 infection, with these symptoms lasting for at least 2 months with no other explanation (https://www.who.int/europe/news-room/fact-sheets/item/post-covid-19-condition). The authors can explain the reasons the participants were at least one month into the recovery period were eligible

• The authors need to describe more clearly how to collect 150 participants out of 798 patients, and the process of collecting data.

• For the Brief-COPE, EQ-5D-5L, and EQ-VAS questionnaires, how to translate them into local languages and validate

• The method part was quite long and had unnecessary repetition. Could the author please shorten it?

• The abstract should not exceed 300 words

Reviewer #2: Great Title, well written manuscript from the abstract, introduction, methodology, statistical analysis, results, discussion till conclusion , Thanks for letting me review this great work, Thanks to the authors

6. PLOS authors have the option to publish the peer review history of their article (what does this mean?). If published, this will include your full peer review and any attached files.

Reviewer #1: No

Reviewer #2: **Yes: **Hani Aiash MD.PhD

Assistant Dean, Associate professor, State University of NY, Upstate Medical University. USA

---

## [Author Response · Author response to Decision Letter 0]

27 Aug 2024

The Editor

PLOS ONE

Dear Editor,

Response to Reviewers: Coping Strategies of COVID-19 recovered patients at the Ghana Infectious Disease Centre

We are grateful to the editor and reviewers for their feedback and helpful recommendations. We are glad to have the chance to resubmit an improved version of our paper for your further evaluation.

In the following, we address the reviewers’ comments and indicate the relevant modifications we applied to the paper. The line numbers we refer to are from the revised manuscript with the track changes off.

Editor’s comments:

We noted in your submission details that a portion of your manuscript may have been presented or published elsewhere. “This is the second manuscript in line with our study on quality of life and coping strategies of COVID-19 recovered patients in Ghana. We have submitted revisions to the first paper i.e. quality of life to this same journal and it is under review. However, the analysis conducted here is not a duplication of the previous paper. We only used some of the results or concepts, particularly quality of life to determine the relationship between coping strategies and quality of life.”

Response: This is the first and only submission of this manuscript anywhere, which forms part of a bigger study on Assessment of The Quality of Life and Coping Strategies of COVID-19 Recovered Patients at The Ghana Infectious Disease Centre (GIDC). The content and issues presented for this manuscript is unique to this publication although the study population remains the same. We have included this explanation in the cover letter. Also, the study on quality of life of COVID-19 recovered patients has recently been published in this journal. https://journals.plos.org/plosone/article?id=10.1371/journal.pone.0306118

Reviewer #1:

1. Post COVID-19 condition, commonly known as long COVID is defined as the continuation or development of new symptoms 3 months after the initial SARS-CoV-2 infection, with these symptoms lasting for at least 2 months with no other explanation (https://www.who.int/europe/news-room/fact-sheets/item/post-covid-19-condition). The authors can explain the reasons the participants were at least one month into the recovery period were eligible.

Response: The study aimed to examine patients who had recovered from COVID-19 at the GIDC and were subsequently discharged for treatment at the post-recovery clinic, where eligibility for inclusion in the study was having spent at least one month there. The focus was not exclusively on those with long-COVID, although some of these patients would have been part of the eligible population.

2. The authors need to describe more clearly how to collect 150 participants out of 798 patients, and the process of collecting data.

Response: Our study population and sampling strategy has been described in the methods section. Kindly refer to lines 162-169 of the revised manuscript. 

3. For the Brief-COPE, EQ-5D-5L, and EQ-VAS questionnaires, how to translate them into local languages and validate.

Response: The official language in Ghana is English which is taught from the basic levels of education. The participants in the study all had some form of formal education and so the study tool was well understood by all, which was facilitated by the researchers.

4. The method part was quite long and had unnecessary repetition. Could the author please shorten it?

Response: We have read over the methods session and made necessary revisions notifying the repetitions and removing them from the text. 

5. The abstract should not exceed 300 words

Response: The abstract has been revised to suit the requirements. (Ref to line numbers 21 – 49)

Reviewer #2:

1. Great title, well written manuscript from the abstract, introduction, methodology, statistical analysis, results, discussion till conclusion. Thanks for letting me review this great work, Thanks to the authors

Response: We thank the reviewer for the feedback.

Thank you

---

## [Decision Letter · Decision Letter 1]

10 Sep 2024

Coping Strategies of COVID-19 recovered patients at the Ghana Infectious Disease Centre

PONE-D-24-14288R1

Dear Dr. Aryeetey,

We’re pleased to inform you that your manuscript has been judged scientifically suitable for publication and will be formally accepted for publication once it meets all outstanding technical requirements.

Kind regards,

Anthony Mwinilanaa Tampah-Naah

Academic Editor

PLOS ONE

Additional Editor Comments (optional):

Reviewers' comments:

Reviewer's Responses to Questions

**Comments to the Author**

1. If the authors have adequately addressed your comments raised in a previous round of review and you feel that this manuscript is now acceptable for publication, you may indicate that here to bypass the “Comments to the Author” section, enter your conflict of interest statement in the “Confidential to Editor” section, and submit your "Accept" recommendation.

Reviewer #1: All comments have been addressed

2. Is the manuscript technically sound, and do the data support the conclusions?

Reviewer #1: Yes

3. Has the statistical analysis been performed appropriately and rigorously? 

Reviewer #1: Yes

4. Have the authors made all data underlying the findings in their manuscript fully available?

Reviewer #1: Yes

5. Is the manuscript presented in an intelligible fashion and written in standard English?

Reviewer #1: Yes

6. Review Comments to the Author

Reviewer #1: Dear authors,

Thank you for the revised version. After reviewing, the corrections made have been effective.

7. PLOS authors have the option to publish the peer review history of their article (what does this mean?). If published, this will include your full peer review and any attached files.

Reviewer #1: No

---

## [Editor Report · Acceptance letter]

8 Nov 2024

PONE-D-24-14288R1 

PLOS ONE

Dear Dr. Aryeetey, 

I'm pleased to inform you that your manuscript has been deemed suitable for publication in PLOS ONE. Congratulations! Your manuscript is now being handed over to our production team.

Kind regards, 

on behalf of

Dr. Anthony Mwinilanaa Tampah-Naah 

Academic Editor

PLOS ONE